# Heat-fueled enzymatic cascade for selective oxyfunctionalization of hydrocarbons

Jaeho Yoon[1,4], Hanhwi Jang [1,4], Min-Wook Oh [2], Thomas Hilberath [3], Frank Hollmann [3], Yeon Sik Jung [1✉] & Chan Beum Park [1✉]

Heat is a fundamental feedstock, where more than 80% of global energy comes from fossil-based heating process. However, it is mostly wasted due to a lack of proper techniques of utilizing the low-quality waste heat (<100 °C). Here we report thermoelectrobiocatalytic chemical conversion systems for heat-fueled, enzyme-catalyzed oxyfunctionalization reactions. Thermoelectric bismuth telluride ($Bi_2Te_3$) directly converts low-temperature waste heat into chemical energy in the form of $H_2O_2$ near room temperature. The streamlined reaction scheme (e.g., water, heat, enzyme, and thermoelectric material) promotes enantio- and chemo-selective hydroxylation and epoxidation of representative substrates (e.g., ethylbenzene, propylbenzene, tetralin, cyclohexane, cis-β-methylstyrene), achieving a maximum total turnover number of rAaeUPO ($TTN_{rAaeUPO}$) over 32000. Direct conversion of vehicle exhaust heat into the enantiopure enzymatic product with a rate of 231.4 µM h$^{-1}$ during urban driving envisions the practical feasibility of thermoelectrobiocatalysis.

[1] Department of Materials Science and Engineering, Korea Advanced Institute of Science and Technology (KAIST), 291 Daehak-ro, Daejeon 34141, Republic of Korea. [2] Department of Materials Science and Engineering, Hanbat National University (HBNU), 125 Dongseodae-ro, Daejeon 34158, Republic of Korea. [3] Department of Biotechnology, Delft University of Technology, Van der Maasweg 9, Delft 2629HZ, The Netherlands. [4]These authors contributed equally: Jaeho Yoon, Hanhwi Jang. ✉email: ysjung@kaist.ac.kr; parkcb@kaist.ac.kr

Energy conversion is a fundamental process promoting transportation, industrial processes, and commercial and residential activities. Today, the primary energy source is still fossil-based with the known challenges for the global climate[1–3]. This issue is further increased by the fact that in all processes primary energy is wasted as off-heat[4]. More than 70% of the global primary energy consumption is lost as waste heat.

High-quality (i.e., high temperature) heat can readily be utilized industrially[5], but waste heat of temperatures below 100 °C still represents a challenge due to the poor Carnot efficiency of the energy conversion processes. In fact, almost half of the global primary energy (45%) is wasted as low-temperature or low-grade waste heat.

The Seebeck effect, the evolution of an electromotive force at conducting materials experiencing a temperature gradient[6], represents a promising approach to valorize low-quality heat wastes[7,8]. Thermoelectric (TE) materials are capable of converting heat into electrical energy based on the Seebeck effect. Bismuth telluride ($Bi_2Te_3$), for example, is a low-temperature thermoelectric material, exhibiting superb dimensionless thermoelectric figure of merit near room temperature[9].

While TE materials have mainly focused on generation of electrical energy previously, the versatile conversion of low-grade heat to chemical energy, which is a substantially more stable form of energy storage compared with electricity, has not been explored yet. This study exploits the capability of $Bi_2Te_3$ to catalyze the oxygen reduction reaction (ORR) through conversion of thermal energy into $H_2O_2$[10] as the first step of our synthetic reaction cascade (Fig. 1). $H_2O_2$ is a versatile reagent in organic synthesis with diverse application fields such as synthetic chemistry, pharmaceutical, electronics, and food industries[11]. Especially, if used with $H_2O_2$-dependent enzymes such as peroxygenases[12,13], various types of stereoselective oxyfunctionalization reactions can be enabled. Hence, employing the thermoelectric properties of $Bi_2Te_3$ to promote peroxygenase-catalyzed oxidation reactions may represent a highly feasible approach to valorize low-temperature waste heat for the synthesis of value-added chemical compounds.

As model peroxygenase to prove the key concept of this study, we selected the recombinant, evolved peroxygenase from *Agrocybe aegerita* (r*Aae*UPO, IUBMB classification: EC 1.11.2.1)[14–16]. r*Aae*UPO catalyzes a range of highly stereoselective hydroxylation and epoxidation reactions converting simple organic compounds into value-added synthetic building blocks for fine chemical and pharmaceutical intermediate synthesis under mild reaction conditions. Combining thermal-induced and $Bi_2Te_3$-catalyzed $H_2O_2$ generation to r*Aae*UPO-catalyzed $H_2O_2$-dependent oxyfunctionalization reactions, we envisioned a platform technology to valorize low-quality heat waste into value-added chemical intermediates.

## Results

### Synthesis and characterization of thermoelectric $Bi_2Te_3$ particles.

We prepared polycrystalline $Bi_2Te_3$ particles via solid-state-synthesis following the procedures reported previously[17]. Low-magnification transmission electron microscopy (TEM) analysis of the particles showed a hexagonal morphology with a lateral size of few micrometers (Fig. 2a). The selected area electron diffraction (SAED) pattern of the particle indicated a polycrystalline morphology of the particles with a hexagonal unit cell (Fig. 2a, inset). The powder X-ray diffraction (XRD) diffractogram in Fig. 2b corroborates that $Bi_2Te_3$ crystallized in a trigonal structure (space group *R-3m*) without detectable secondary phase. In addition, we confirmed that the $Bi_2Te_3$ was a narrow-gap semiconductor with a bandgap of 0.14 eV (Supplementary Fig. 1).

We further analyzed the chemical homogeneity and composition of the synthesized $Bi_2Te_3$. Figure 2c shows a scanning transmission electron microscopy (STEM) image acquired by a high-angle annular dark-field (HAADF) detector. As HAADF imaging is sensitive to atomic number (Z) difference, the uniform contrast in Fig. 2c indicates a very homogeneous distribution of Bi and Te in the particle. This is in good agreement with the energy-dispersive X-ray spectroscopy (EDX) elemental mapping results, showing no evidence for precipitations in the particles. The EDX quantification results indicate that the atomic fraction of Te is slightly higher than expected from the molecular formula $Bi_2Te_3$. $Bi_2Te_3$ synthesized under Te-rich condition exhibits an n-type conduction property owing to donor-like $Te_{Bi}$ antisite defects[9]. Indeed, the measured Seebeck coefficient (S) of $Bi_2Te_3$ at 308 K was ~ −148.1 µV K$^{-1}$ (Fig. 2d), close to the literature[18]. The negative sign of S implies that the $Bi_2Te_3$ is an n-type semiconductor. Hall measurements at room temperature also showed that the electron concentration of $Bi_2Te_3$ was ~3 × 10$^{19}$ cm$^{-3}$ (Supplementary Table 1). Furthermore, the magnitude of S slightly increased with increasing temperature, suggesting that $Bi_2Te_3$ is a degenerate semiconductor with n-type conduction properties (Supplementary Fig. 2)[6]. Overall, the negative signs of both Seebeck coefficients and Hall coefficients confirmed that the synthesized $Bi_2Te_3$ was a degenerate n-type semiconductor.

### Fueling peroxygenase catalysis via thermoelectrocatalytic in situ $H_2O_2$ generation.

We scrutinized the capability of as-synthesized $Bi_2Te_3$ particles for thermoelectrocatalytic reduction of $O_2$ to $H_2O_2$ under applied temperature difference ($\Delta T$) using a homebuilt reactor (Supplementary Fig. 3). Note that the particles were constantly stirred inside the reactor to prevent reaching thermal equilibrium ($\Delta T \approx 0$). The homogeneous $Bi_2Te_3$ solution (5 mg mL$^{-1}$) with $\Delta T$ of 45 K generated $H_2O_2$ with a maximum rate of 0.051 ± 0.005 mM h$^{-1}$ under ambient $O_2$ atmosphere, whereas $H_2O_2$ was not observed under $N_2$-enriched condition or in the absence of $Bi_2Te_3$ or in thermally equilibrated solutions (Supplementary Fig. 4, Fig. 3a, b). We confirmed a linear relationship between the amount of thermoelectrocatalytically generated $H_2O_2$ and $\Delta T$ without a noticeable saturation (Fig. 3a), implying a direct correlation between catalytic activity of TE materials and $\Delta T$ (and Seebeck potential, $V = -S \Delta T$) vide infra.

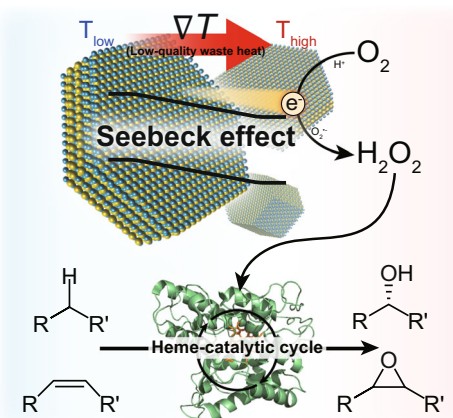

**Fig. 1 Schematic illustration of a thermoelectrobiocatalytic $Bi_2Te_3$/ r*Aae*UPO cascade for selective oxyfunctionalization reactions.** Thermoelectric $Bi_2Te_3$ particles drive oxygen reduction reaction to generate $H_2O_2$ with applied temperature difference. Unspecific peroxygenase catalyzes various selective aerobic oxygenation reactions by utilizing in situ generated $H_2O_2$.

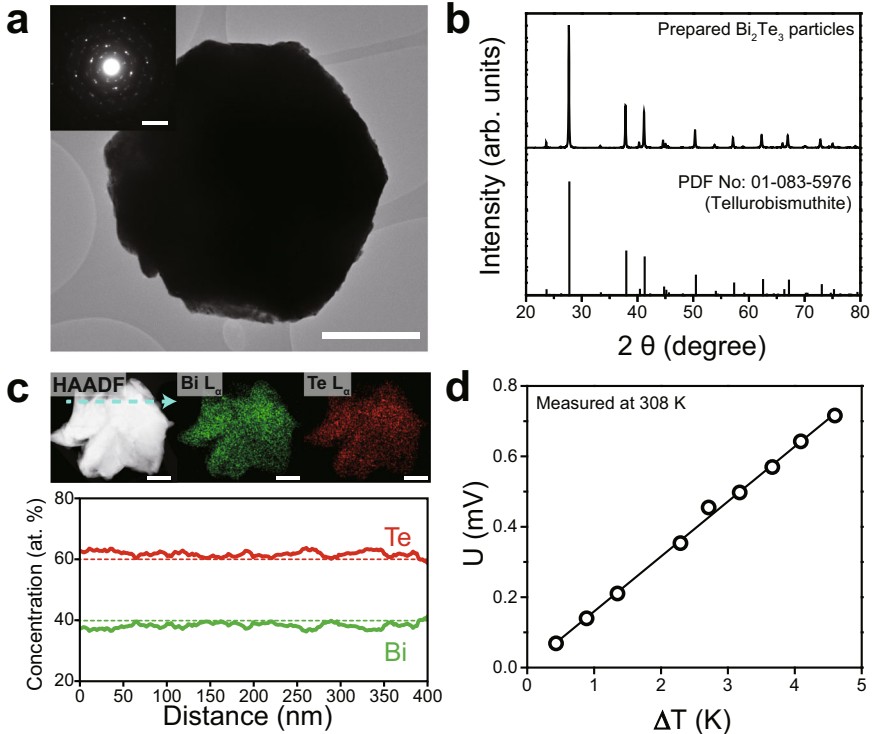

**Fig. 2 Material characterization of the as-synthesized $Bi_2Te_3$ particles. a** Low-magnification transmission electron microscopy (TEM) image of the synthesized $Bi_2Te_3$ particle. Scale bar = 500 nm. Inset is the corresponding SAED pattern. Inset scale bar = 5 $nm^{-1}$. **b** Powder XRD pattern of the synthesized $Bi_2Te_3$. Scan rate: 5° $min^{-1}$. **c** HAADF-STEM image, EDS elemental mapping results, and STEM-EDS quantification results of dashed line. Green and red dashed lines denote the stoichiometric concentration of Bi and Te in $Bi_2Te_3$, respectively. Scale bar = 100 nm. **d** Measured electrical potential versus temperature gradient of the synthesized $Bi_2Te_3$ at 308 K. The positive slope of U vs. $\Delta T$ indicates the negative sign of the relative Seebeck coefficient.

We investigated the possibility of fueling peroxygenase-catalyzed hydroxylation reactions by the incessant $H_2O_2$ generation from $Bi_2Te_3$ under an applied $\Delta T$. Crude r$Aae$UPO without further purification after cultivation was used instead of purified enzyme, which is more practical and merits for industrialization. We used 5 mg $mL^{-1}$ $Bi_2Te_3$ because $H_2O_2$ accumulation saturated at ~0.03 mM (Fig. 3a, *vide infra*) beyond the $Bi_2Te_3$ concentration. As shown in Fig. 3c, the $Bi_2Te_3$/UPO couple catalyzed the selective conversion of ethylbenzene to (*R*)-1-phenylethanol. Applying a temperature gradient of 45 K for two days resulted in the formation of 325.4 μM of enantiopure (*R*)-product (>99% *ee*) with no acetophenone overoxidation product detectable. Additional experiments revealed that only the positive control group produced (*R*)-1-phenylethanol of 178.3 ± 22.7 μM (>99% *ee*, reaction time: 2 h), verifying that heat-fueled $H_2O_2$ generation of $Bi_2Te_3$ is crucial for thermoelectrobiocatalytic ethylbenzene hydroxylation (Fig. 3d). The initial product formation rate in these experiments (0.122 mM $h^{-1}$, Fig. 3c) significantly exceeded the $H_2O_2$ formation rate determined above (0.051 mM $h^{-1}$). We attribute this observation to the irreversible peroxygenase step removing $H_2O_2$ from the steady-state equilibrium.

To demonstrate general applicability, we tested a variety of enantioselective oxyfunctionalization reactions, including benzylic hydroxylation of propylbezene (**4–6**) and tetralin (**7–9**), hydroxylation of cyclohexane (**10–12**) and epoxidation of *cis*-β-methylstyrene (**13–14**)] (Fig. 4). For ethylbenzene hydroxylation, maximum $TTN_{rAaeUPO}$ of 10,770 was achieved with high selectivity (>99% *ee*) using 25 nM r$Aae$UPO. The $Bi_2Te_3$/UPO catalyzed the selective oxygenation reactions of non-activated $sp^3$ C–H bonds, recording the highest $TTN_{rAaeUPO}$ over 32,000 with dominant (*R*)-products

(propylbenzene: $TTN_{rAaeUPO}$ of 32,913, 74.1% *ee*) within 48 h. Enantiopure oxygenated products were also obtained from the enzymatic conversion of tetralin ($TTN_{rAaeUPO}$ of 6361, >99% *ee*), cyclohexane ($TTN_{rAaeUPO}$ of 10,559), and *cis*-β-methylstyrene ($TTN_{rAaeUPO}$ of 18,095, >99% *ee*). Again, control experiments revealed that thermoelectrocatalysis (TEC) was essential to supply the peroxygenase-reactions (Supplementary Fig. 5).

**Effect of thermoelectric parameters on thermoelectrobiocatalysis.** Recent reports substantiated the influence of thermoelectric effect on catalysis, where the generation of Seebeck potential within TE materials (as a catalyst support or promoter) alters the electron work functions of catalysts, resulting in so-called thermoelectric promotion of catalysis (TEPOC)[19,20]. We hypothesized that thermoelectrobiocatalysis (TEBC), where the TE material (i.e., $Bi_2Te_3$) plays a role as a stand-alone catalyst itself, could be also promoted by the Seebeck effect based on the TEPOC model. In principle, a dependency of reaction rate on the Seebeck potential in TEPOC can be expressed as follows [Eq. (1)]:

$$Ln(r/r_0) = -\gamma \cdot S \cdot \Delta T/k_b T_h \qquad (1)$$

where $r$ is the reaction rate, $r_0$ is the reaction rate when $\Delta T$ is zero, $\gamma$ is an empirically determined dimensionless constant, $k_b$ is the Boltzmann constant, and $T_h$ is the temperature at hot side. Based on the generalized relationship between the catalytic activity and the thermoelectric characteristics, we systematically scrutinized the effect of each thermoelectric parameters (i.e., $\Delta T$ and $S$) on the thermo-electrocatalytic $H_2O_2$ generation and the thermoelectrobiocatalytic ethylbenzene hydroxylation reaction, respectively.

We examined $H_2O_2$ generation capacity of $Bi_2Te_3$ with respect to $\Delta T$. As described before, we observed that the $H_2O_2$

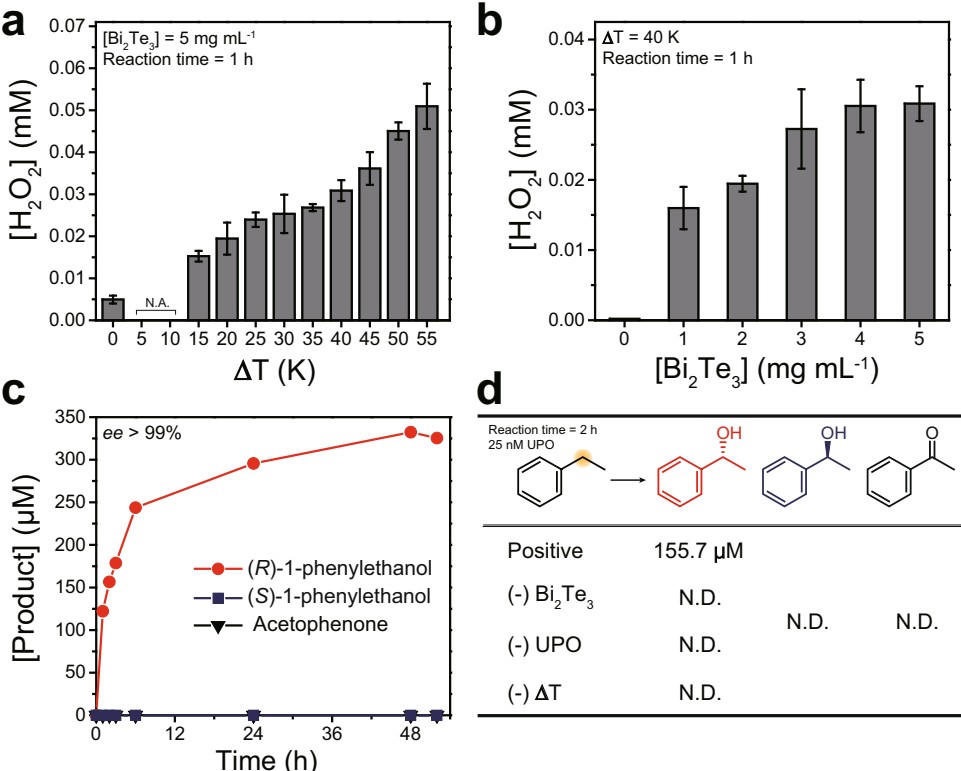

**Fig. 3 Thermoelectrocatalytic H₂O₂ generation and thermoelectrobiocatalytic ethylbenzene hydroxylation using thermoelectric Bi₂Te₃ particles.**
**a** Amount of generated $H_2O_2$ with increasing $\Delta T$. Reaction conditions: 5 mg mL⁻¹ Bi₂Te₃ dispersed in an O₂-purged KPB (100 mM, pH 7.0) with applied temperature difference. **b** Amount of generated $H_2O_2$ with increasing concentration of Bi₂Te₃. Reaction conditions: Bi₂Te₃ dispersed in an O₂-purged KPB (100 mM, pH 7.0) with applied $\Delta T$ (45 K). All reported values represent the mean ± standard deviation ($n = 3$). **c** A time course of thermoelectrobiocatalytic conversion of ethylbenzene to (R)-1-phenylethanol, (S)-1-phenylethanol, and acetophenone for 50-h reaction. **d** A series of control experiments for each reaction components for 2-h reaction. Reaction conditions: 5 mg mL⁻¹ Bi₂Te₃, 200 nM rAaeUPO, and 100 mM ethylbenzene dispersed in an O₂-purged KPB (100 mM, pH 7.0) with applied $\Delta T$ (45 K). For control experiments, 25 nM rAaeUPO was used instead. Note that all quantities were determined from gas chromatographic analyses. The negative sign (−) denotes that the component is excluded as a control experiment. N.A. not applicable, N.D. not detected.

production rate increases with increasing $\Delta T$ (Fig. 3b), because larger $\Delta T$ induces a higher Seebeck potential that tunes the Fermi level (or electron chemical potential) to further energetically favorable position. Plotting the initial $H_2O_2$ generation rates as a function of $\Delta T$ (Fig. 5a) revealed two linear regions with $\gamma$ values of 11.22 and 8.12 [Eq. (1)]. The linear dependency of Ln $(r/r_0)$ on $\Delta T$ indicates that thermoelectrocatalytic $H_2O_2$ formation is indeed facilitated by the TEPOC effect, corroborating the role of Bi₂Te₃ itself as a TEPOC center for boosting $H_2O_2$ generation by a self-generated TEPOC effect. The reduction of $\gamma$ at $\Delta T$ above 25 K is ascribed to the accelerated decomposition of $H_2O_2$ at high temperatures (Supplementary Fig. 6). This diminution of $H_2O_2$ was ascertained to be a thermoelectrocatalytic process through additional control experiments (Supplementary Fig. 7).

We conducted ultraviolet photoelectron spectroscopy (UPS) measurements to understand the energetics of TEC. The Fermi level ($E_F$) and the valence band maximum (VBM) of the Bi₂Te₃ were −4.3 and −4.54 eV, respectively, where the vacuum level is set to 0 eV (Supplementary Fig. 8)[21,22]. The conduction band minimum (CBM) is located at −4.4 eV with respect to the vacuum level, given that the optical bandgap of Bi₂Te₃ is 0.14 eV (Supplementary Fig. 1); therefore, the $E_F$ is located 0.1 eV above the CBM, which is consistent to the degenerate n-type characteristics of the synthesized Bi₂Te₃. The potential of VBM, CBM, and Fermi level are 0.04, −0.1, and −0.2 V (vs. NHE), respectively (Supplementary Fig. 9). These results present that the potential of CBM and the Fermi level of Bi₂Te₃ are more negative

than the redox potential of $H_2O_2/H_2O$ (1.76 V vs. NHE[23]) or $H_2O_2/OH^•$ (0.87 V vs. NHE[24]), further underpinning that thermoelectrocatalytic reduction of $H_2O_2$ is thermodynamically favorable[10,25]. The further reduction of $H_2O_2$ in the reaction mixture has been also reported in other rAaeUPO-driven catalytic systems[26,27].

The TEPOC effect resulted in an enhanced rate of the Bi₂Te₃/rAaeUPO-driven theromobiocatalytic oxyfunctionalization reaction. As shown in Fig. 5b, the ethylbenzene-to-(R)-1-phenylethanol conversion rate and the product yield were each doubled from $1.3 \pm 0.4$ to $2.8 \pm 0.5$ μM min⁻¹ and $107.6 \pm 20.8$ to $252.8 \pm 16.3$ μM with the increasing $\Delta T$ from 15 to 45 K. This observation can be ascribed to both, increasing catalytic activity of the biocatalyst with temperature (Supplementary Figs. 10 and 11) as well as the increasing $H_2O_2$-generation rate. Thermal inactivation of rAaeUPO can be excluded as the temperature range investigated here was well in the acceptable temperature range for rAaeUPO[15].

We further analyzed the catalytic promotion effect of $S$ on TEBC. The $S$ values of Bi₂Te₃ were mechanically controlled via high-energy ball milling (without introducing dopants) to exclude any contribution of chemical composition change affecting the reaction routes. Such mechanical deformation induces a donor-like effect in Bi₂Te₃ by forming Te vacancies[28]; thus, we obtained samples with smaller $S$ values by increasing the ball milling time ($S$ of −150.5 μV K⁻¹ for as-spun samples; −109.6 μV K⁻¹ for 6 h ball-milled samples; −97.4 μV K⁻¹ for 24 h ball-milled samples,

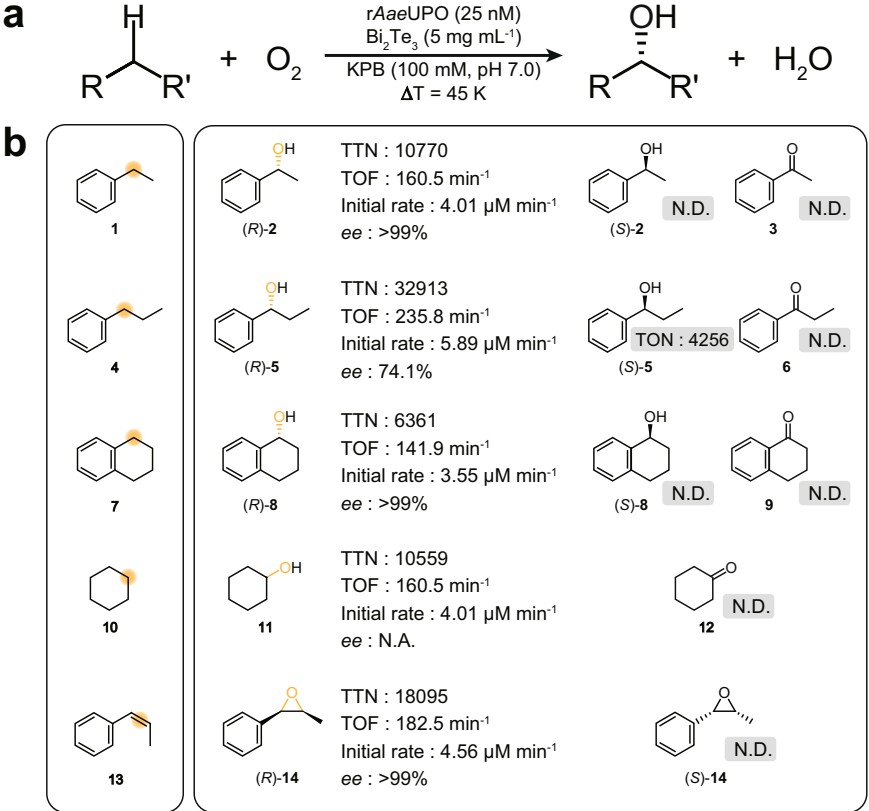

**Fig. 4 Substrate scope of thermoelectrobiocatalytic oxyfunctionalization reactions by Bi₂Te₃/rAaeUPO. a** A brief reaction equation of the thermoelectrobiocatalytic oxyfunctionalization of hydrocarbons by the Bi₂Te₃/UPO couple. **b** Selective hydroxylation reactions of ethylbezene (**1**), propylbenzene (**4**), cyclohexane (**7**), and tetralin (**10**), and selective styrene epoxidation reaction of *cis*-β-methylstyrene (**13**). Note that all quantities were determined from gas chromatographic analyses. Initial rate and TOF were determined at 15 min of reaction (30 min for hydroxylation of cyclohexane). TTN and *ee* were determined by the maximum value of reactions. Reaction conditions: 5 mg mL$^{-1}$ Bi₂Te₃, 25 nM rAaeUPO, and 100 mM ethylbenzene dispersed in an O₂-purged KPB (100 mM, pH 7.0) with applied Δ$T$ (45 K). For styrene epoxidation of compound **13**, we applied 35 K of Δ$T$ to avoid the ignition of compound **13**. Reaction time = 48 h. N.D. not detected, N.A. not applicable.

Supplementary Fig. 12). Note that the change of $S$ value of Bi₂Te₃ was negligible (~10%) when heated up to 100 °C. As shown in Supplementary Fig. 13, we found that H₂O₂ production rate increased with the increasing $S$. This phenomenon is similar to the effect of Δ$T$ because the increase of both parameters should result in higher Seebeck potential enhancing the reaction rate. Figure 5c shows the dependency of H₂O₂ generation on $S$ by a linear plot with the fitted constant $\gamma$ of 6, indicating the TEPOC effect of sole Bi₂Te₃. Accordingly, the ethylbenzene hydroxylation rate escalated (ethylbenzene conversion rate: 2.5 ± 0.8 to 2.8 ± 0.5 μM min$^{-1}$; product yield: 238.9 ± 30.5 to 325.1 ± 18.8 μM; >99% *ee*; reaction time: 24 h) as $S$ values rose from −88.6 to −150.5 μV K$^{-1}$ (Fig. 5d), corroborating the thermoelectric promotion effect on the Bi₂Te₃/rAaeUPO system. We ascribed the inconsistency between the uptrend of H₂O₂ generation and ethylbenzene conversion to the formation/decomposition behavior of Compound I during UPO catalytic cycle, which complexifies the overall kinetics of Bi₂Te₃/UPO-driven selective oxyfunctionalization reactions[29].

**Thermoelectrobiocatalytic recycle of exhaust heat from urban driving.** In modern society, transportation generates ~20% of total low-grade waste heat around the world; for example, over 30% of the fuel energy is lost as an exhaust gas in conventional vehicles[4,30]. Therefore, exhaust gas of vehicle is a great source of low-quality heat waste to hand, as well as recovering waste heat from exhaust gas of vehicles would be highly desirable for

environmental sustainability. Typical waste heat of vehicle is converted into electricity by a massive TE generator (TEG) consisting of several TE modules[31]. However, the current TEG-based exhaust waste heat recovery is not an attractive way to provide sufficient economic and environmental benefits to the society due to the poor efficiency and limited power output of the TEGs.

We envision that the thermoelectrobiocatalytic exhaust heat recovery system (Fig. 6a) can relieve the aforementioned dilemma of exhaust heat utilization, and also provide a creative way of utilizing dumped waste heat into value-added fine chemicals with much greener and economical way for upscale reactors in industrial complexes. The temperature of a rear muffler of vehicles reaches ~100 °C under normal operation; thus, the rear muffler of the vehicle is a suitable part where the accessibility and the temperature are beneficial for low-grade waste heat-driven thermoelectrobiocatalytic systems. To explore the feasibility of TEBC on the real-life application, we attached a thermoenzymatic reactor to the rear muffler part (Fig. 6a, right panel and 6b), and examined the production of (R)-1-phenylethanol from ethylbenzene hydroxylation under typical urban driving conditions. Figure 6b shows a spatial temperature distribution of the rear muffler and the attached cylindrical glass reactor. The inside temperature of the rear muffler and the reactor was measured to 2 °C and 20.3 °C, respectively. We set the driving route between the Korea Advanced Institute of Science and Technology (KAIST) and Hanbat National University (HBNU) in Daejeon, Korea, with a round-trip distance of ~15 km (Fig. 6c and Supplementary Fig. 14). After 30 min urban driving with a maximum

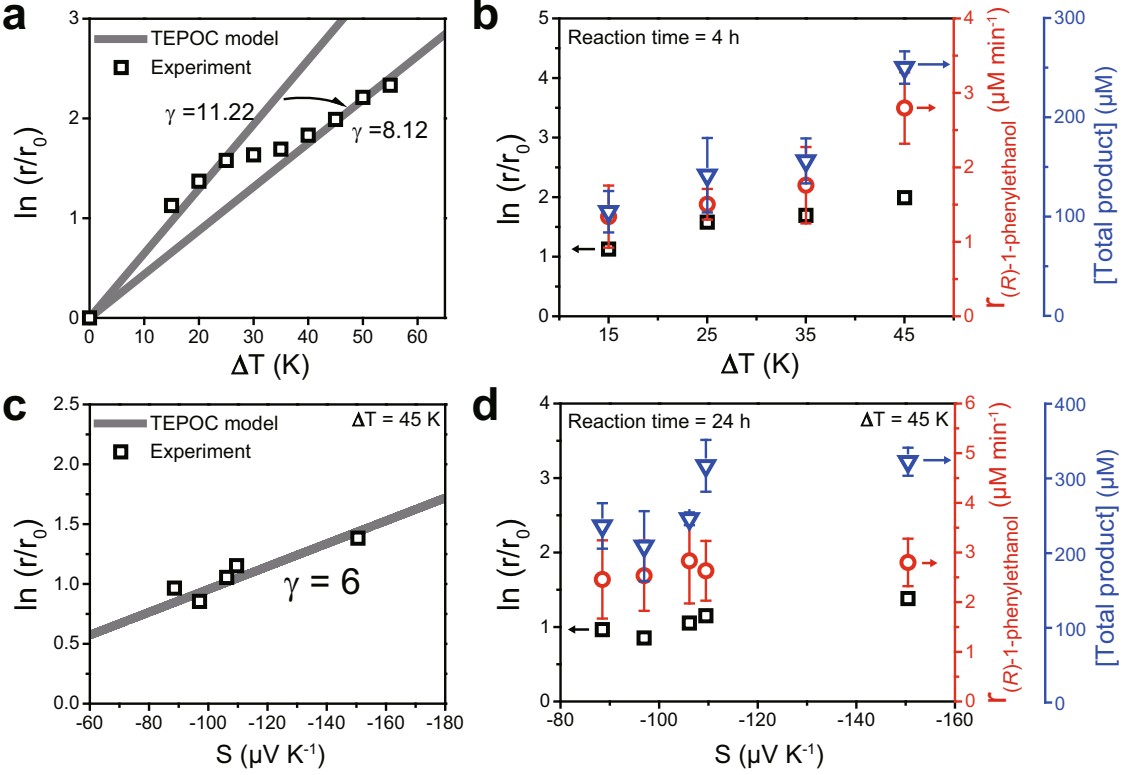

**Fig. 5 Effect of thermoelectric parameters on catalytic activity of thermoelectrocatalytic $H_2O_2$ generation and thermoelectrobiocatalytic ethylbenzene hydroxylation.** The dependencies of $H_2O_2$ production rate, (R)-1-phenylethanol production rate, and product yield of thermoelectrobiocatalytic ethylbenzene conversion on (**a**, **b**) applied $\Delta T$ from 0 to 55 K and (**c**, **d**) Seebeck coefficient (S) of the $Bi_2Te_3$, respectively. Reaction conditions: 5 mg mL$^{-1}$ $Bi_2Te_3$ dispersed in an $O_2$-purged KPB (100 mM, pH 7.0) with applied $\Delta T$. For biocatalytic reactions, 50 nM rAaeUPO and 100 mM ethylbenzene were also dispersed in an $O_2$-purged KPB (100 mM, pH 7.0). $H_2O_2$ production rates were determined at 60 min of reaction for (**a**, **b**) and 15 min of reaction for (**c**, **d**). Ethylbenzene conversion rates were determined at 60 min of reaction. Product yield was determined at 4 h of reaction for (**b**) and 24 h of reaction for (**d**).

speed of 50 km h$^{-1}$, the muffler's temperature reached to ~97.8 °C (Fig. 6b). Consequently, the average temperature of the glass reactor increased from 20.3 °C to 31.5 °C. The large temperature difference between the muffler and the glass reactor is ascribed to the high thermal resistance between them and the low thermal conductivity of the glass reactor. Nevertheless, the convection of air during the driving continuously cooled off the reactor, thus providing a sufficient temperature gradient in the reactor for $Bi_2Te_3$/rAaeUPO-driven thermoelectrobiocatalytic reactions. As shown in Fig. 6d, the $Bi_2Te_3$/rAaeUPO system successfully converted exhaust heat into enantiopure (R)-1-phenylethanol with average rate of 231.4 µM h$^{-1}$ (275.4, 132.0, and 294.8 µM h$^{-1}$ for each drive) after thrice of urban driving. Note that negligible production was observed during idle states, supporting the generation of pure oxygenated products by the exhaust heat-recycling thermoelectrobiocatalytic process.

## Discussion

The direct conversion of low-grade waste heat into valuable oxyfunctionalized chemicals with high selectivity highlights the potential of TEBC in sustainable and renewable energy applications. Admittedly, TEBC itself is in its infancy and yet opposes challenges against practical utilizations that must be circumvented. $Bi_2Te_3$ exhibited relatively low $H_2O_2$ production rate (<0.05 mM h$^{-1}$) that consequentially limits the total product yields (<1 mM). Other representative TE materials (e.g., Se-doped $Bi_2Te_3$[32], $Cu_2Se$[33], and SnSe[34]) also showed low $H_2O_2$ production rate (<0.03 mM h$^{-1}$, see Supplementary Fig. 15), which we attribute to the weak driving force (i.e., Seebeck potential) with few mV (<10 mV) scale at low-temperature region (<100 °C).

Energetically preferred thermoelectrocatalytic reduction of $H_2O_2$ further lowers the overall yields. Besides, the use of poorly water-soluble organic substrates and products having high volatility in the hot-water reaction media can arise the loss of net product yield.

Thermoelectrocatalytically generated reactive oxygen species (i.e., $O_2^{\bullet-}$ and $OH^{\bullet}$, Supplementary Fig. 16) can impede the biocatalyst stability because they inflict severe oxidative stress to the enzyme, particularly by oxidatively degrading the catalytic heme prosthetic group[13,27,35]. Furthermore, we found an irreversible decrease of Te-to-Bi ratio at the surface of $Bi_2Te_3$ from 1.667 (Supplementary Fig. 17a, point 1) to 1.156 (Supplementary Fig. 17a, point 2) after 8 h of thermoelectrobiocatalytic reactions. The STEM-EDS elemental mapping confirmed that the loss of Te was mainly due to the incorporation of oxygen into Te sites (Supplementary Fig. 17), resulting the formation of bismuth oxide at the surface.

We could reveal that the electrons in the conduction band react with $O_2$ during our thermoelectrocatalytic ORR, which explains the main reaction for $H_2O_2$ production; however, the other half-reaction to sustain the whole reaction would need more clarification. From the energy band diagram, the VBM of the $Bi_2Te_3$ (0.04 V vs NHE) is not sufficiently positive to oxidize water ($H_2O/O_2$, 1.23 V vs. NHE[36]) or hydroxide ($OH^-/OH^{\bullet}$, 1.99 V vs. NHE[37]). Therefore, it is energetically unfavorable for holes to govern the other half-reaction. The use of typical electron donors (e.g., methanol and formaldehyde) did not boost the $H_2O_2$ productivity of $Bi_2Te_3$ as well (Supplementary Fig. 18).

Alternatively, we speculate that the consumed electrons in $Bi_2Te_3$ can be replenished by accommodating oxygen into $Bi_2Te_3$

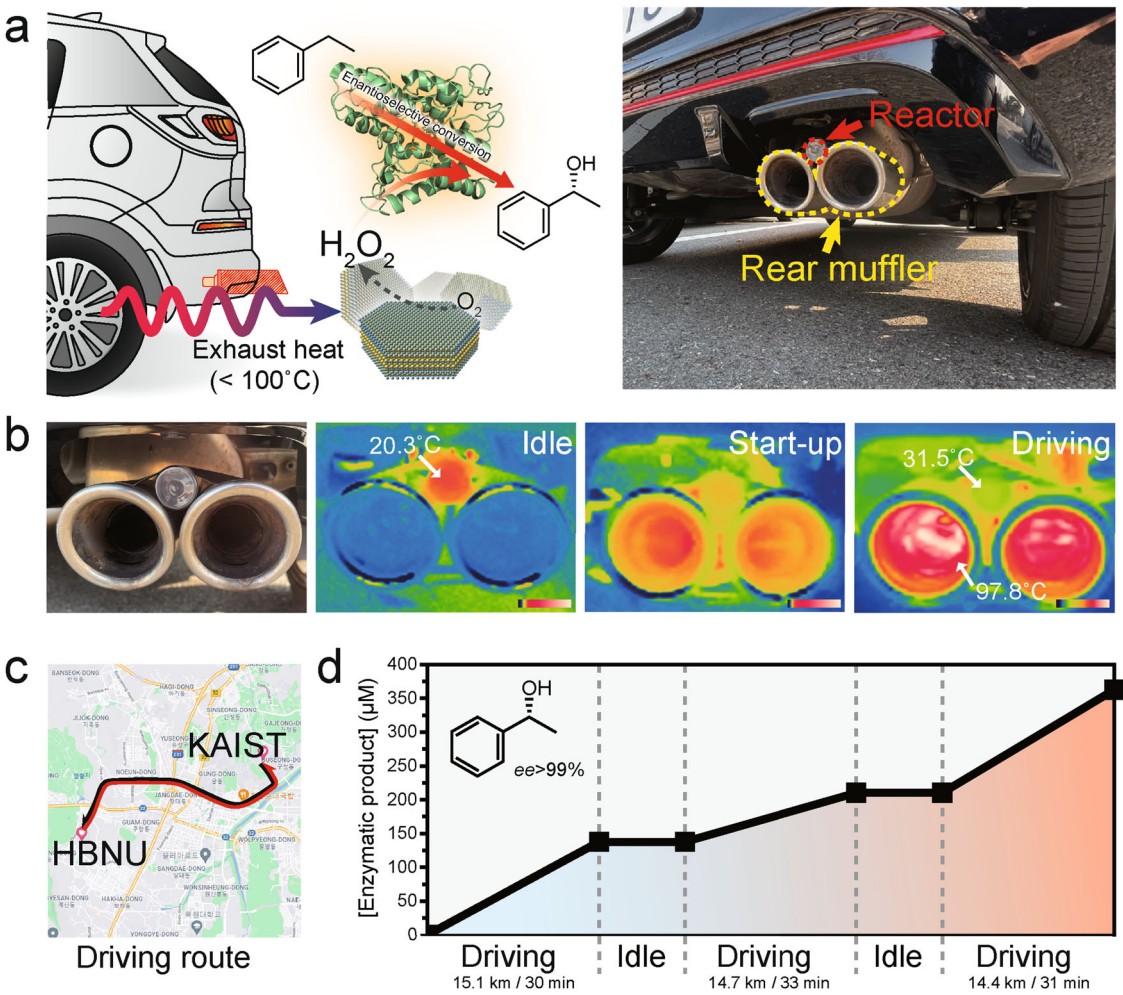

**Fig. 6 Waste heat mining for fine chemical synthesis during urban driving. a** A scheme and a digital photograph of the practical application of TEBC for vehicle exhaust-heat recovery system. A glass reactor was attached on the rear muffler of the vehicle. **b** Digital photographs of the vehicle rear muffler before and after driving, which was recorded by thermal imaging infrared camera. Color bars denote the measured temperature from 0 to 100 °C on a linear scale. **c** Selected driving route between Korea Advanced Institute of Science and Technology (KAIST) and Hanbat National University (HBNU). Map data: ©2022 Google, TMAP Mobility. **d** Generation of enantiopure (*R*)-1-phenylethanol using the exhaust-heat-fueled Bi$_2$Te$_3$/UPO system during urban driving. Reaction conditions: 5 mg mL$^{-1}$ Bi$_2$Te$_3$, 1 μM r*Aae*UPO, and 100 mM ethylbenzene dispersed in 20 mL O$_2$-purged KPB (100 mM, pH 7.0).

lattice, thus acting as the other half-reaction for ORR. It is known that the incorporated oxygen in Bi$_2$Te$_3$ acts as an electron donor by a complex interaction between vacancy, oxygen, and antisite defects to maintain charge neutrality of the matrix[38,39]. Similar cases have been widely reported in photocatalytic reactions, where the catalyst itself participates in the half-reaction[40–42].

Nevertheless, the Bi$_2$Te$_3$/UPO couple operates well as an efficient heat-fueled biocatalytic platform, displaying noteworthy activity toward a variety of hydrocarbon oxyfunctionalization reactions with high selectivity. The highest TTN$_{r*Aae*UPO}$ of the TEBC measured in this study is comparable to, or even exceeds that of other state-of-the-art r*Aae*UPO-driven biocatalytic systems (Supplementary Fig. 19). We found that the overall catalytic activity was mainly boosted by TEPOC effect of Bi$_2$Te$_3$, where the thermoelectrocatalytic reaction rate was well fitted by a function of *S*. The demonstration of TEPOC effect in terms of *S* by a mechanical deformation methodology (i.e., high-energy ball milling) not only proves the underlying fundamental of thermopower-driven catalysis but also signifies the importance of *S* than other physical properties such as particle size (Supplementary Fig. 20) and crystallinity (Supplementary Figs. 21 and 22) that varies during high-energy ball milling process. This

signposts future efforts for progressing TEBC to a practical level that the aforesaid bottleneck—deficient supply of H$_2$O$_2$—should be addressed by maximizing the Seebeck potential of TE materials for given reaction conditions. Therefore, TE materials with a colossal *S* value must take priority for enhancing TEBC performance[43,44]. For example, it is known that the maximum *S* value of Bi$_2$Te$_3$ can be as high as −250 μV K$^{-1}$ via careful tuning of the Fermi level[9]. Therefore, based on the TEPOC model, we expect that the best achievable H$_2$O$_2$ generation rate of Bi$_2$Te$_3$ might be 10.9 times faster (~4.03 μM min$^{-1}$, when $r_0$ is 0.37 μM min$^{-1}$) than that without the TEPOC effect if we utilize Bi$_2$Te$_3$ particles with a *S* value of −250 μV K$^{-1}$. Moreover, we expect band engineering would be an ideal strategy to synergistically improve the performance of the reaction. It is known that the band edge positions of both valence band and conduction band can be tuned by alloying Se to Te sites[45]. Together with tuning carrier concentration, one might achieve an optimal alignment of the Fermi level and reaction potential. Furthermore, TEBC has a room for further engineering betterment of the system, such as surface engineering of catalyst[46] that can improve the H$_2$O$_2$ generation capacity or a new reactor design[47] that can address the substrate/product loss issue and maximize the energy

conversion efficiency. A methodology to generate a large temperature gradient to the reactor while consuming less power than that of current convection-based reactor would eventually increase the overall conversion efficiency of the TEBC. Thermal interface material to the junction between the reactor and the heat source or a radiation shield around the reactor might be introduced to achieve this.

TEBC is now come up on stage. The omnipresence of heat in our lives and the universality of catalytic mechanism of TEBC not only grant high efficiency but also guarantee superior compatibility to preexisting energy applications (e.g., vehicle internal combustion engine), which makes TEBC a "flexible" system.

This work substantiates the potential of combining TE materials and biocatalysis for versatile conversion of low-grade heat waste to value-added chemical intermediates. Through theoretical and experimental analyses, we demonstrated the $Bi_2Te_3$/rAaeUPO couple-driven selective oxyfunctionalization reactions of hydrocarbons, and unveiled the direct relationship between the thermoelectric parameters and catalytic activity, which can be explained by the TEPOC model. The demonstration of TEBC gives significant benefits over traditional approaches for driving peroxygenase-catalyzed chemical conversions. For example, the $O_2$-reduction pathway without artificial electron donors in aqueous environment provides more atom-efficient peroxide generation route. The simple reaction scheme without ancillary cocatalysts or mediators promotes high purity of target chemical feedstocks, making TEBC a much greener and environmentally benign process. Furthermore, the ubiquity of thermal energy enables interminable driving of selective oxyfunctionalization reactions unlike choppy sunlight, elevating TEBC to more practical and economical level. TEBC proved its real-life applicability beyond a proposal of a new concept of catalysis by utilizing vehicle exhaust heat as a driving force for $Bi_2Te_3$/rAaeUPO catalyzed (R)-1-phenylethanol production, envisioning much greener and economical application of waste heat. To sum up, TEBC paves a creative way to exploit low-grade waste heat for green fine chemical synthesis, developing a sustainable waste heat recovery technique.

## Methods

**$Bi_2Te_3$ synthesis**. Thermoelectric $Bi_2Te_3$ particles were synthesized by ball milling and solid-state reaction method. High-purity raw elements of Bi shots (5N plus, 99.999%) and Te shots (5N plus, 99.999%) were used without any purification. Stoichiometric amounts of elements and zirconia balls with a diameter of 10 mm were weighed and put into the zirconia jar. To prevent oxidation during the milling process, the jar was evacuated, filled with $N_2$ gas, and sealed inside the glovebox. The elements were pulverized by planetary ball milling apparatus (Pulverisette 7, Fritsch) with a rotational speed of 500 rpm for 2 h. The obtained powder was cold-pressed by the hydraulic press under a uniaxial pressure of 200 MPa. Then, the pellet was annealed in a tube furnace at 723 K for 12 h under an Ar atmosphere. The annealed pellet was hand-ground by agate mortar for 10 min, and the powder was sieved under 25 μm to remove agglomerated particles.

**$Bi_2Te_3$ characterization**. The phase purity of the synthesized $Bi_2Te_3$ was examined using powder X-ray diffractometer (SmartLab, Rigaku, Japan). A Cu $K_\alpha$ radiation source having a characteristic wavelength of 0.15406 nm was used under an operating voltage of 45 kV and a current of 200 mA. The diffractogram was acquired for $2\theta$ angle ranging from 20° to 80° with a scan rate of 5° min$^{-1}$. High-angle annular dark-field (HAADF)-STEM imaging and selected area electron diffraction analysis were conducted using a Talos F200X transmission electron microscope (FEI Company, USA). Energy-dispersive X-ray spectroscopy (EDX) signals were collected using a Super X EDS detector (ChemiSTEM technology, USA) integrated into the TEM apparatus. The Seebeck coefficient of the synthesized $Bi_2Te_3$ was measured by a commercial apparatus (ZEM-3, ULVAC-RIKO, Japan). The sample was cold-pressed into a cylindrical shape and machined into a rectangular bar shape ($3 \times 3 \times 10$ mm$^3$). The measurement was conducted with a weak He backpressure to improve contacts between the sample and the probe, and to minimize possible outgassing at high temperatures. The temperature difference of the sample was controlled to be within 10 K, because excess temperature gradient of the sample will result in inhomogeneity of $S$ along with the sample. Note

that $Bi_2Te_3$ powder was not sintered to prevent the change of TE properties during the sintering process.

**Manipulating Seebeck coefficient of $Bi_2Te_3$**. The Seebeck coefficient of $Bi_2Te_3$ was mechanically adjusted using identical planetary ball milling apparatus. The purpose of this mechanical deformation was to generate excess electrons without introducing extrinsic dopants (e.g., halogens in Te sites or alkaline earth metals in Bi sites) which can affect the overall reaction route. $Bi_2Te_3$ particles were put into the zirconia jar together with zirconia balls with a diameter of 10 mm. The jar was also evacuated and sealed inside the glovebox to prevent oxidation due to excessive heat during milling. The ball-to-powder ratio was 10:1, and the rotational speed was 500 rpm. We found that powders were prone to stick to the jar wall, which may result in uneven milling of particles. Therefore, the jar wall was scratched with a tool to detach the powders from the wall. We performed additional ball milling experiments with the rotational speed of 300 rpm for fine-tuning of $S$ values. Other conditions such as ball-to-powder ratio or the diameter of ball remained unchanged.

**Diffuse reflectance infrared Fourier transform spectroscopy (DRIFTS)**. Diffuse reflectance spectra of $Bi_2Te_3$ powders were measured using an iS50 spectrometer (Thermo Fisher, USA). The absorption coefficient was calculated from the reflectance by the Kubelka–Munk equation: $F(R) = (1-R)^2/2R$, where $R$ is the optical reflectance.

**Hall effect measurement**. The Hall coefficient ($R_H$) of $Bi_2Te_3$ was measured using HMS-8407 Hall effect measurement system (Lake Shore Cryotronics, USA). The pelletized $Bi_2Te_3$ was electrically connected to the apparatus using sharp nickel probes with a van der Pauw configuration. The magnitude of an applied magnetic field, and excitation current was 1 T and 20 mA, respectively. The effect of misalignment voltage was eliminated by measuring Hall resistance under reversed field direction. The Hall carrier concentration was calculated by $n_H = 1/eR_H$, where $e$ is the elementary charge.

**Ultraviolet photoelectron spectroscopy (UPS)**. UPS spectra were measured using electron spectroscopy for chemical analysis (ESCA) apparatus (Axis Supra, Kratos, UK) under an ultrahigh vacuum (base pressure $<5 \times 10^{-10}$ Torr). He I radiation source with a photon energy of 21.2 eV was used. The apparatus was calibrated using a gold standard sample stored in a load lock chamber. The sample surface was etched using Ar cluster sputter gun for 60 s before acquisition to exclude the effect of surface oxidation because the typical information depth of UPS is ~2–3 nm. The Fermi level ($E_F$) in the unit of eV with respect to the vacuum level ($E_{vac}$) was determined using following equation:

$$E_{vac} - E_F = -h\nu + E_{cutoff} \tag{2}$$

where $h\nu$ is the energy of incident light (21.2 eV) and $E_{cutoff}$ is the secondary electron cut-off energy.

**Reactive oxygen species (ROS) detection**. The amounts of superoxide ions ($O_2^{\bullet-}$) and hydroxyl radicals ($OH^\bullet$) were estimated using the nitro blue tetrazolium (NBT) assay and terephthalic acid (TA) assay, respectively[48]. We added $Bi_2Te_3$ and reagents ([NBT] = 20 μM, [TA] = 300 μM) to KPB (100 mM, pH 7.0), and applied temperature gradient to the solution. For NBT assays, a change in the sample's absorbance at 259 nm was monitored using a V-650 spectrophotometer (JASCO Inc., Japan) after reactions. The fluorescence intensity of the sample solution was recorded at 430 nm ($\lambda_{ex} = 315$ nm) using an FP6500 spectrofluorometer (JASCO Inc., Japan) for TA assays.

**$H_2O_2$ quantification**. The as-synthesized n-type $Bi_2Te_3$ particles were dispersed in a potassium phosphate-buffered solution (KPB, 100 mM, pH 7.0; $O_2$, $N_2$, or air purged) with applied temperature difference. Note that gas was continuously provided into the solution through a Teflon tube (diameter: 1 mm) during the reaction. The temperature at the cold side of the reactor ($T_c$) is fixed to 10 °C. The concentration of $H_2O_2$ was quantified by a colorimetric assay using horseradish peroxidase (HRP)-catalyzed oxidation of ABTS. 50 μL of reaction sample was added to 950 μL of the reagent solution [1 mM ABTS and 2.5 U HRP dissolved in KPB (100 mM, pH 5.0)], and incubated for 5 min at room temperature. The incubated sample was centrifuged for 1.5 min at $21{,}200 \times g$ to remove $Bi_2Te_3$ particles, and the supernatant was collected. Then, we monitored the absorbance of the collected sample at 420 nm using a V-650 spectrophotometer (JASCO Inc., Japan).

**Enzyme preparation**. A recombinant unspecific peroxygenase from *Agrocybe aegerita* (rAaeUPO) was obtained via recombinant expression in *Pichia pastoris* according to a previous method[16]. A UPO stock solution with a green color was used without any purification steps. The specific activity of prepared rAaeUPO was estimated to be $176.9 \pm 20.9$ U mg$^{-1}$. One unit of enzyme activity was defined as the amount of enzyme catalyzing the oxidation of 1 μmol of ABTS per minute in a potassium phosphate-buffered solution (KPB, 100 mM, pH 5.0).

**Enzyme activity assay**. The enzymatic activity of rAaeUPO was evaluated using the peroxygenase activity assay with 2,2'-azino-bis(3-ethylbenzothiazoline-6-sulfonic acid) (ABTS) molecules as substrate. A reaction sample was mixed with a reagent solution (v/v, 50:950); the reagent solution was prepared by dissolving 0.5 mM $H_2O_2$, and 0.5 mM ABTS in a potassium phosphate-buffered solution (100 mM, pH 5.0). The oxidation of ABTS was monitored by the change in the absorption intensity at 420 nm using a V-650 spectrophotometer (JASCO Inc., Japan) and activities were calculated using a Beer-Lambert law. Note that the molar extinction coefficient of ABTS at 420 nm ($\varepsilon_{420}$) is 36.0 mM$^{-1}$ cm$^{-1}$.

**Thermoelectrobiocatalytic reaction and analysis**. Fixed concentrations of as-synthesized n-type $Bi_2Te_3$, rAaeUPO, and 100 mM of model substrates (e.g., ethylbenzene, propylbenzene, cyclohexane, tetralin, and cis-β-methylstyrene) were dispersed in an $O_2$-purged KPB (100 mM, pH 7.0). Poorly soluble organic substrates were first dispersed in a buffered solution and vigorously stirred for 1–2 min to achieve a homogeneous dispersion. Note that oxygen gas was continuously provided into the solution through a Teflon tube (diameter: 1 mm) during the reactions. The temperature at the cold side of the reactor ($T_c$) is fixed to 10 °C and the temperature at the hot side of the reactor ($T_h$) is regulated by a heater. The amount of enzymatic products was quantified using a 7890A gas chromatography (Agilent Technologies Inc., USA) equipped with a CP-Chirasil-Dex CB column (25 m, 0.32 mm, 0.25 μm). Specifically, 50 μL of a reaction sample was collected, and the organic products in the sample were extracted using ethyl acetate (containing 5 mM 1-octanol as the internal standard). After 5 min of vigorous stirring, the sample was centrifuged at $21,200 \times g$ for 1.5 min. Then, the organic phase was carefully collected and dried with $MgSO_4$, followed by additional centrifugation at $21,200 \times g$ for 1.5 min. The oven temperature programs were tabulated in Supplementary Table 2. The total turnover number (TTN), turnover frequency (TOF), and enantiomeric excess (ee) of rAaeUPO were calculated according to the Eqs. (3), (4), and (5) as follows:

$$\text{TTN}_{rAaeUPO} = \text{Maximum concentration of product/concentration of } rAaeUPO \quad (3)$$

$$\text{TOF}_{rAaeUPO} = \text{Turnover number of } rAaeUPO/\text{Time} \quad (4)$$

$$ee = [\text{moles of enantiomer} - \text{moles of other enantiomer}]/\text{total moles of product} \cdot 100 \quad (5)$$

**Urban driving test**. $Bi_2Te_3$ (5 mg mL$^{-1}$), rAaeUPO (1 μM), and ethylbenzene were dispersed in 20 mL of $O_2$-purged KPB (100 mM, pH 7.0). The reaction solution was put into cylindrical glass vial (total volume of 20 mL) and completely sealed using a Parafilm® M sealing film. The glass reactor was attached to the rear muffler of a vehicle (Morning 3rd generation, KIA, Korea). The speed and route during each driving were monitored using a mobile navigation application (TMAP, SK telecom, Korea). The amount of enzymatic products was quantified using a 7890A gas chromatography after each driving. The temperature of the rear muffler and the reactor was measured using a thermal imaging infrared camera.

## Data availability

The data that support the findings of this study are available within the main text of this article and its Supplementary Information. Any other relevant data are available from the corresponding authors upon request.

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

## Acknowledgements

This work was supported by the National Research Foundation (NRF) of Korea via the Creative Research Initiative Center (grant no. NRF-2015R1A3A2066191), Republic of Korea.

## Author contributions

J.Y. and H.J. conceived the research, performed experiments, analyzed the data, and wrote the manuscript. H.J. and M.-W.O. measured and analyzed thermoelectric properties of thermoelectric materials. T.H. and F.H. synthesized the biocatalyst. Y.S.J. and C.B.P. directed the whole project and reviewed the manuscript. All authors participated in the discussion of the results and approved the manuscript.

## Competing interests

The authors declare no competing interests.
