## [Peer Review File · Nature Communications]

Thermoelectrobiocatalysis: Heat-fueled enzymatic cascade for selective oxyfunctionalization of hydrocarbonsREVIEWER COMMENTS

Reviewer #1 (Remarks to the Author):

This manuscript reports the use of thermoelectric effect for the improved reaction activity of a few selective oxyfunctionalization of hydrocarbons. Together with the reference 10, it is a novel application of thermoelectric effect for the conversion of low grade waste heat to useful chemical feedstocks. This is also the first independent confirmation of the rate equation 1 (The equation 1, first proposed in the references 19 & 20) which relates the catalytic activity increase directly with the Seebeck voltage. The carefully designed control experiment results (without TE material, or ΔT , or UPO), and the results on the dependence on the Seebeck Coefficient S from the ball milling samples, give strong support to the TEPOC explanation for the observed phenomena.

Generally speaking the manuscript is well structured and well written. The work is novel and the results are very interesting. All the conclusions are supported well by the experimental evidence, although it is not ideal to have 3 points only in the Figure 4C. Why isn't the r/r_0 for Figure 4C in log scale whilst Figure 4a is?

For the Fig. 2 caption, it is better to explain the exact meaning of (-) Bi₂Te₃ and (-)UPO and (-) ΔT . Although this is explained as "without" in the main content, it is still better to make it clear in the figure caption for the benefit of easy reading.

I think "thermoelectrocatalysis" and "thermoelectrobiocatalysis" are better than "thermocatalysis" and "thermobiocatalysis" respectively, as the key here is the temperature gradient (thermoelectric), not the temperature (thermal) itself.

Reviewer #2 (Remarks to the Author):

This is an interesting paper that reported thermobiocatalytic chemical conversion using thermoelectric Bi₂Te₃ to form H₂O₂. However, there are still some missing points in the article, which need to be addressed.

1. Author did not mention that the used Bi₂Te₃ is n or p-type. It looks like an n-type material according to Fig 1D and Fig S2. It should write a clear explanation in the experimental part.

2. A energy diagram for Bi_2Te_3 and its related catalytic reaction should be provided; a detailed mechanism is better to be provided as well.
3. Because this research is in its infancy stage, I suggest making predictions about the best achievable performance, for example, the energy levels could be tuned in Bi_2Te_3 .
4. It would be great to compare this TEPOC with other related technologies, even if current productivity is still low.

Answers to Referees' Comments

Title: Thermoelectrobiocatalysis: Heat-fueled enzymatic cascade for selective oxyfunctionalization of hydrocarbons

Authors: Jaeho Yoon, Hanhwi Jang, Min-Wook Oh, Thomas Hilberath, Frank Hollmann, Yeon Sik Jung*, and Chan Beum Park*

First of all, we would like to thank the Referees for having taken the time to deliver valuable comments on our manuscript. Following is a point-by-point response and a summary of our revision made according to the comments.

Referee #1

This manuscript reports the use of thermoelectric effect for the improved reaction activity of a few selective oxyfunctionalization of hydrocarbons. Together with the reference 10, it is a novel application of thermoelectric effect for the conversion of low grade waste heat to useful chemical feedstocks. This is also the first independent confirmation of the rate equation 1 (The equation 1, first proposed in the references 19 & 20) which relates the catalytic activity increase directly with the Seebeck voltage. The carefully designed control experiment results (without TE material, or ΔT , or UPO), and the results on the dependence on the Seebeck Coefficient S from the ball milling samples, give strong support to the TEPOC explanation for the observed phenomena.

1. Generally speaking the manuscript is well structured and well written. The work is novel and the results are very interesting. All the conclusions are supported well by the experimental evidence, although it is not ideal to have 3 points only in the Figure 4C.

Answer: We appreciate the Referee's valuable comment. We have conducted additional experiments and included the data in the **Figures 5C** and **5D** in the revised manuscript. We have revised the original manuscript accordingly.

[Manuscript, page 8, line 20 – page 9, line 4]

Fig. 5C shows the dependency of H_2O_2 generation on S by a **linear** plot with the fitted constant γ of 6, indicating the TEPOC effect of sole Bi_2Te_3 . Accordingly, the ethylbenzene hydroxylation rate escalated (ethylbenzene conversion rate: 2.5 ± 0.8 to $2.8 \pm 0.5 \mu M \text{ min}^{-1}$; product yield: 238.9 ± 30.5 to $325.1 \pm 18.8 \mu M$; $>99\%$ *ee*; reaction time: 24 h) as S values rose from -88.6 to $-150.5 \mu V \text{ K}^{-1}$ (**Fig. 5D**), corroborating the thermoelectric promotion effect on the $Bi_2Te_3/tAaeUPO$ system. We ascribed the inconsistency between the uptrend of H_2O_2 generation rate ($>50\%$) and ethylbenzene conversion rate ($\sim 10\%$) to the formation/decomposition behavior of Compound I during UPO catalytic cycle, which complexifies the overall kinetics of Bi_2Te_3/UPO -driven selective oxyfunctionalization reactions.²⁹

[Manuscript, page 16, lines 7-9]

We performed additional ball milling experiments with the rotational speed of 300 rpm for fine-tuning of S values. Other conditions such as ball-to-powder ratio or the diameter of ball remained unchanged.

Fig. 5| Effect of thermoelectric parameters on catalytic activity of thermoelectrocatalytic H_2O_2 generation and thermoelectrobiocatalytic ethylbenzene hydroxylation. The dependencies of H_2O_2 production rate, (*R*)-1-phenylethanol production rate, and product yield of thermoelectrobiocatalytic ethylbenzene conversion on (A, B) applied ΔT from 0 to 55 K and (C, D) Seebeck coefficient (*S*) of the Bi_2Te_3 , respectively. Reaction conditions: 5 mg mL^{-1} Bi_2Te_3 dispersed in an O_2 -purged KPB (100 mM, pH 7.0) with applied ΔT . For biocatalytic reactions, 50 nM *rAaeUPO* and 100 mM ethylbenzene were also dispersed in an O_2 -purged KPB (100 mM, pH 7.0). H_2O_2 production rates were determined at 60 min of reaction for (A, B) and 15 min of reaction for (C, D). Ethylbenzene conversion rates were determined at 60 min of reaction. Product yield was determined at 4 h of reaction for (B) and 24 h of reaction for (D).

2. Why isn't the r/r_0 for Figure 4C in log scale whilst Figure 4A is?

Answer: We appreciate this kind comment. As shown in comment #1, we have modified the y-axis of **Figure 5C** in log scale in the revised manuscript.

3. For the Fig. 2 caption, it is better to explain the exact meaning of (-) Bi_2Te_3 and (-) UPO and (-) ΔT . Although this is explained as "without" in the main content, it is still better to make it clear in the figure caption for the benefit of easy reading.

Answer: In response to the comment, we have clarified the meaning of negative sign (-) in the captions of the Figures (**Fig. 3** and **Supplementary Figures 5, 7, and 16**) in the revised manuscript. We have revised the original manuscript accordingly.

[Caption of Fig. 3]

Note that all quantities were determined from gas chromatographic analyses. **The negative sign (-) denotes that the component is excluded as a control experiment.** N.A. = not applicable. N.D. = not detected.

[Caption of Supplementary Figure 5]

For styrene epoxidation of *cis*- β -methylstyrene, we applied ΔT of 35 K to avoid the ignition of *cis*- β -methylstyrene. Reaction time = 2 h. **The negative sign (-) denotes that the component is excluded as a control experiment.** N.D. = not detected.

[Caption of Supplementary Figure 7]

Note that the initial concentration of H₂O₂ in solution was 1 mM. The negative sign (-) denotes that the component is excluded as a control experiment.

[Caption of Supplementary Figure 16]

All reported values represent the mean ± standard deviation (n = 3). The negative sign (-) denotes that the component is excluded as a control experiment.

4. I think “thermoelectrocatalysis” and “thermoelectrobiocatalysis” are better than “thermocatalysis” and “thermobiocatalysis” respectively, as the key here is the temperature gradient (thermoelectric), not the temperature (thermal) itself.

Answer: We agree with this comment and thus replaced thermocatalysis and thermobiocatalysis with thermoelectrocatalysis (TEC) and thermoelectrobiocatalysis (TEBC), respectively, throughout the revised manuscript.

[Manuscript, page 1, title]

Thermo**electro**biocatalysis: Heat-fueled enzymatic cascade for selective oxyfunctionalization of hydrocarbons

[Manuscript, page 1, lines 20-21]

Here we report thermo**electro**biocatalytic chemical conversion systems for heat-fueled, enzyme-catalyzed oxyfunctionalization reactions.

[Manuscript, page 1, lines 27-29]

Direct conversion of vehicle exhaust heat into the enantiopure enzymatic product with a rate of 231.4 μM h⁻¹ during urban driving envisions the practical feasibility of thermo**electro**biocatalysis.

[Manuscript, page 4, line 19]

Fueling peroxygenase catalysis via thermo**electro**catalytic in situ H₂O₂ generation

[Manuscript, page 4, lines 20-22]

We scrutinized the capability of as-synthesized Bi₂Te₃ particles for thermo**electro**catalytic reduction of O₂ to H₂O₂ under applied temperature difference (ΔT) using a homebuilt reactor (**Supplementary Fig. 3**).

[Manuscript, page 5, lines 2-5]

We confirmed a linear relationship between the amount of thermo**electro**catalytically generated H₂O₂ and ΔT without a noticeable saturation (**Fig. 3A**), implying a direct correlation between catalytic activity of TE materials and ΔT (and Seebeck potential, $V = -S\Delta T$) vide infra.

[Manuscript, page 5, lines 14-17]

Additional experiments revealed that only the positive control group produced (*R*)-1-phenylethanol of 178.3±22.7 μM (>99% *ee*, reaction time: 2 h), verifying that heat-fueled H₂O₂ generation of Bi₂Te₃ is crucial for thermo**electro**biocatalytic ethylbenzene hydroxylation (**Fig. 3D**).

[Manuscript, page 6, lines 6-8]

Again, control experiments revealed that thermo**electro**catalysis (**TEC**) was essential to supply the peroxygenase-reactions (**Supplementary Fig. 5**).

[Manuscript, page 6, line 10]

Effect of thermoelectric parameters on thermo**electro**biocatalysis

[Manuscript, page 6, lines 14-16]

We hypothesized that thermoelectrobiocatalysis (TEBC), where the TE material (i.e., Bi₂Te₃) plays a role as a stand-alone catalyst itself, could be also promoted by the Seebeck effect based on the TEPOC model.

[Manuscript, page 6, lines 21-24]

Based on the generalized relationship between the catalytic activity and the thermoelectric characteristics, we systematically scrutinized the effect of each thermoelectric parameters (i.e., ΔT and S) on the thermoelectrocatalytic H₂O₂ generation and the thermoelectrobiocatalytic ethylbenzene hydroxylation reaction, respectively.

[Manuscript, page 7, lines 6-8]

The linear dependency of $\ln(r/r_0)$ on ΔT indicates that thermoelectrocatalytic H₂O₂ formation is indeed facilitated by the TEPOC effect, corroborating the role of Bi₂Te₃ itself as a TEPOC center for boosting H₂O₂ generation by a self-generated TEPOC effect.

[Manuscript, page 7, lines 10-12]

This diminution of H₂O₂ was ascertained to be a thermoelectrocatalytic process through additional control experiments (**Supplementary Fig. 7**).

[Manuscript, page 8, line 10]

We further analyzed the catalytic promotion effect of S on TEBC.

[Manuscript, page 9, line 6]

Thermoelectrobiocatalytic recycle of exhaust heat from urban driving

[Manuscript, page 9, line 16 – page 10, line 1]

We envision that the thermoelectrobiocatalytic exhaust heat recovery system (**Fig. 6A**) can relieve the aforementioned dilemma of exhaust heat utilization, and also provide a creative way of utilizing dumped waste heat into value-added fine chemicals with much greener and economical way for upscale reactors in industrial complexes. The temperature of a rear muffler of vehicles reaches approximately 100°C under normal operation; thus, the rear muffler of the vehicle is a suitable part where the accessibility and the temperature are beneficial for low-grade waste heat-driven thermoelectrobiocatalytic systems. To explore the feasibility of TEBC on the real-life application, we attached a thermoenzymatic reactor to the rear muffler part (**Figs. 6A, right panel and 6B**), and examined the production of (*R*)-1-phenylethanol from ethylbenzene hydroxylation under typical urban driving conditions.

[Manuscript, page 10, lines 11-13]

Nevertheless, the convection of air during the driving continuously cooled off the reactor, thus providing a sufficient temperature gradient in the reactor for Bi₂Te₃/*rAae*UPO-driven thermoelectrobiocatalytic reactions.

[Manuscript, page 10, lines 16-18]

Note that negligible production was observed during idle states, supporting the generation of pure oxygenated products by the exhaust heat-recycling thermoelectrobiocatalytic process.

[Manuscript, page 10, lines 21-24]

The direct conversion of low-grade waste heat into valuable oxyfunctionalized chemicals with high selectivity highlights the potential of TEBC in sustainable and renewable energy applications. Admittedly, TEBC itself is in its infancy and yet opposes challenges against practical utilizations that must be circumvented.

[Manuscript, page 11, lines 5-6]

Energetically preferred thermoelectrocatalytic reduction of H₂O₂ further lowers the overall yields.

[Manuscript, page 11, lines 9-14]

Thermoelectrocatalytically generated reactive oxygen species (i.e., O₂^{•-} and OH[•], **Supplementary Fig. 16**) can impede the biocatalyst stability because they inflict severe oxidative stress to the enzyme, particularly by oxidatively degrading the catalytic heme prosthetic group.^{13, 27, 35} Furthermore, we found an irreversible decrease of Te-to-Bi ratio at the surface of Bi₂Te₃ from 1.667 (**Supplementary Fig. 17A**, point 1) to 1.156 (**Supplementary Fig. 17A**, point 2) after 8 h of thermoelectrobiocatalytic reactions.

[Manuscript, page 12, lines 11-13]

We found that the overall catalytic activity was mainly boosted by TEPOC effect of Bi₂Te₃, where the thermoelectrocatalytic reaction rate was well fitted by a function of *S*.

[Manuscript, page 12, lines 18-21]

This signposts future efforts for progressing **TEBC** to a practical level that the aforesaid bottleneck—deficient supply of H₂O₂—should be addressed by maximizing the Seebeck potential of TE materials for given reaction conditions. Therefore, TE materials with a colossal *S* value must take priority for enhancing **TEBC** performance.⁴³⁻⁴⁴

[Manuscript, page 13, lines 6-11]

Furthermore, **TEBC** has a room for further engineering betterment of the system, such as surface engineering of catalyst⁴⁶ that can improve the H₂O₂ generation capacity or a new reactor design⁴⁷ that can address the substrate/product loss issue and maximize the energy conversion efficiency. A methodology to generate a large temperature gradient to the reactor while consuming less power than that of current convection-based reactor would eventually increase the overall conversion efficiency of the **TEBC**.

[Manuscript, page 13, lines 14-17]

TEBC is now come up on stage. The omnipresence of heat in our lives and the universality of catalytic mechanism of **TEBC** not only grant high efficiency but also guarantee superior compatibility to preexisting energy applications (e.g., vehicle internal combustion engine), which makes **TEBC** a “flexible” system.

[Manuscript, page 14, lines 1-2]

The demonstration of **TEBC** gives significant benefits over traditional approaches for driving peroxygenase-catalyzed chemical conversions.

[Manuscript, page 14, lines 4-13]

The simple reaction scheme without ancillary co-catalysts or mediators promotes high purity of target chemical feedstocks, making **TEBC** a much greener and environmentally benign process. Furthermore, the ubiquity of thermal energy enables interminable driving of selective oxyfunctionalization reactions unlike choppy sunlight, elevating **TEBC** to more practical and economical level. **TEBC** proved its real-life applicability beyond a proposal of a new concept of catalysis by utilizing vehicle exhaust heat as a driving force for Bi₂Te₃/*rAae*UPO catalyzed (*R*)-1-phenylethanol production, envisioning much greener and economical application of waste heat. To sum up, **TEBC** paves a creative way to exploit low-grade waste heat for green fine chemical synthesis, developing a sustainable waste heat recovery technique.

[Manuscript, page 19, line 3]

Thermoelectrobiocatalytic reaction and analysis. Fixed concentrations of as-synthesized n-type Bi₂Te₃, *rAae*UPO, and 100 mM of model substrates (e.g., ethylbenzene, propylbenzene, cyclohexane, tetralin, and *cis*- β -methylstyrene) were dispersed in an O₂-purged KPB (100 mM, pH 7.0).

[Caption of Fig. 1]

Fig. 1| Schematic illustration of a thermoelectrobiocatalytic Bi₂Te₃/rAaeUPO cascade for selective oxyfunctionalization reactions.

[Caption of Fig. 3]

Fig. 3| Thermoelectrocatalytic H₂O₂ generation and thermoelectrobiocatalytic ethylbenzene hydroxylation using thermoelectric Bi₂Te₃ particles. (A) Amount of generated H₂O₂ with increasing ΔT . Reaction conditions: 5 mg mL⁻¹ Bi₂Te₃ dispersed in an O₂-purged KPB (100 mM, pH 7.0) with applied temperature difference. (B) Amount of generated H₂O₂ with increasing concentration of Bi₂Te₃. Reaction conditions: Bi₂Te₃ dispersed in an O₂-purged KPB (100 mM, pH 7.0) with applied ΔT (45 K). All reported values represent the mean \pm standard deviation (n = 3). (C) A time course of thermoelectrobiocatalytic conversion of ethylbenzene to (*R*)-1-phenylethanol, (*S*)-1-phenylethanol, and acetophenone for 50-h reaction.

[Caption of Fig. 4]

Fig. 4| Substrate scope of thermoelectrobiocatalytic oxyfunctionalization reactions by Bi₂Te₃/rAaeUPO. (A) A brief reaction equation of the thermoelectrobiocatalytic oxyfunctionalization of hydrocarbons by the Bi₂Te₃/UPO couple.

[Caption of Fig. 5]

Fig. 5| Effect of thermoelectric parameters on catalytic activity of thermoelectrocatalytic H₂O₂ generation and thermoelectrobiocatalytic ethylbenzene hydroxylation. The dependencies of H₂O₂ production rate, (*R*)-1-phenylethanol production rate, and product yield of thermoelectrobiocatalytic ethylbenzene conversion on (A, B) applied ΔT from 0 to 55 K and (C, D) Seebeck coefficient (S) of the Bi₂Te₃, respectively.

[Caption of Fig. 6]

(A) A scheme and a digital photograph of the practical application of TEBC for vehicle exhaust-heat recovery system.

[Supporting information, page 1, title]

Thermoelectrobiocatalysis: Heat-fueled enzymatic cascade for selective oxyfunctionalization of hydrocarbons

[Caption of Supplementary Figure 3]

(A) An illustration of thermoelectrobiocatalytic reactor setup with thermocouple equipped heater and refrigerated bath circulator. The reaction solution is continuously mixed using a magnetic stirrer with stirring speed of 200 rpm during reactions. (B) A digital photograph of the homebuilt reactor. The heater temperature was stabilized using PID controller. The thermocouple and heating wire were fixed to the reactor using a ceramic adhesive. (C) Digital photographs recorded by thermal imaging infrared camera during thermoelectrocatalytic reactions.

[Caption of Supplementary Figure 4]

Effect of ambient gas atmosphere on thermoelectrocatalytic generation of H₂O₂.

[Caption of Supplementary Figure 7]

Effect of the omission of each reaction component (i.e., Bi₂Te₃, ΔT) and ambient gas atmosphere on thermoelectrocatalytic decomposition of H₂O₂.

[Caption of Supplementary Figure 7]

Schematic energy diagram of as-synthesized n-type Bi₂Te₃ particles for thermoelectrocatalytic O₂ reduction to H₂O₂.

[Caption of Supplementary Figure 13]

A time course of thermoelectrocatalytic H₂O₂ generation by the high-energy ball milled Bi₂Te₃ particles with different S under O₂ atmosphere for 90 min of reaction.

[Caption of Supplementary Figure 16]

Qualitative analyses of O₂^{•-} and OH[•] generation during thermoelectrocatalytic H₂O₂ generation via a series (A) nitro blue tetrazolium (NBT, 20 μM) and (B) terephthalic acid (TA, 300 μM) assays with the omission of Bi₂Te₃ (5 mg mL⁻¹) and ΔT (45 K), respectively.

[Caption of Supplementary Figure 17]

HAADF-STEM image of Bi₂Te₃ particles after 8 h of thermoelectrocatalytic reactions.

[Caption of Supplementary Figure 18]

Effect of typical electron donors on Bi₂Te₃-driven thermoelectrocatalytic generation of H₂O₂.

Referee #2

This is an interesting paper that reported thermobiocatalytic chemical conversion using thermoelectric Bi₂Te₃ to form H₂O₂. However, there are still some missing points in the article, which need to be addressed.

1. Author did not mention that the used Bi₂Te₃ is n or p-type. It looks like an n-type material according to Fig 1D and Fig S2. It should write a clear explanation in the experimental part.

Answer: We appreciate the Referee's valuable comment. We confirmed the n-type conductivity of Bi₂Te₃ from the negative Seebeck coefficients and Hall coefficients. We have included the relevant explanation in the revised manuscript.

[Manuscript, page 4, lines 16-17]

Overall, the negative signs of both Seebeck coefficients and Hall coefficients confirmed that the synthesized Bi₂Te₃ was a degenerate n-type semiconductor. Overall, these results confirm that the synthesized Bi₂Te₃ was high quality polycrystalline with degenerate n-type TE property.

[Manuscript, page 17, line 22 – page 18, line 1]

H₂O₂ quantification: The as-synthesized n-type Bi₂Te₃ particles were dispersed in a potassium phosphate-buffered solution (KPB, 100 mM, pH 7.0, O₂ or N₂ or air purged) with applied temperature difference.

[Manuscript, page 19, lines 3-6]

Thermoelectrocatalytic reaction and analysis. Fixed concentrations of as-synthesized n-type Bi₂Te₃, rAaeUPO, and 100 mM of model substrates (e.g., ethylbenzene, propylbenzene, cyclohexane, tetralin, and cis- β -methylstyrene) were dispersed in an O₂-purged KPB (100 mM, pH 7.0).

2. An energy diagram for Bi₂Te₃ and its related catalytic reaction should be provided; a detailed mechanism is better to be provided as well.

Answer:

1. Energy diagram

In response to the Referee's recommendation, we have investigated the energetics of thermoelectrocatalysis for our n-type Bi₂Te₃ particles. We conducted ultraviolet photoelectron spectroscopy (UPS) measurements to understand the energetics of TEC. The Fermi level (E_F) and the valence band maximum (VBM) of the Bi₂Te₃ were -4.3 eV and -4.54 eV, respectively, where the vacuum level is set to 0 eV (**Supplementary Fig. 8**). The conduction band minimum (CBM) is located at -4.4 eV with respect to the vacuum level, given that the optical band gap of Bi₂Te₃ is 0.14 eV (**Supplementary Fig. 1**); therefore, the E_F is located 0.1 eV above the CBM, which is consistent with the degenerate n-type characteristics of the synthesized Bi₂Te₃. The potential of VBM, CBM, and Fermi level are 0.04, -0.1, and -0.2 V (vs. NHE), respectively (**Supplementary Fig. 9**), which are consistent with the literature [*Adv. Funct. Mater.* **2017**, *27*, 1701823; *Mater. Adv.* **2020**, *1*, 450-462].

[Manuscript, page 7, line 13 – page 8, line 1]

We conducted ultraviolet photoelectron spectroscopy (UPS) measurements to understand the energetics of TEC. The Fermi level (E_F) and the valence band maximum (VBM) of the Bi₂Te₃ were -4.3 eV and -4.54 eV, respectively, where the vacuum level is set to 0 eV (**Supplementary Fig. 8**).²¹⁻²² The conduction band minimum (CBM) is located at -4.4 eV with respect to the vacuum level, given that the

optical band gap of Bi_2Te_3 is 0.14 eV (**Supplementary Fig. 1**); therefore, the E_F is located 0.1 eV above the CBM, which is consistent to the degenerate n-type characteristics of the synthesized Bi_2Te_3 . The potential of VBM, CBM, and Fermi level are 0.04, -0.1, and -0.2 V (vs. NHE), respectively (**Supplementary Fig. 9**). These results present that the potential of CBM and the Fermi level of Bi_2Te_3 are more negative than the redox potential of $\text{H}_2\text{O}_2/\text{H}_2\text{O}$ (1.76 V vs. NHE²³) or $\text{H}_2\text{O}_2/\text{OH}^-$ (0.87 V vs. NHE²⁴), further underpinning that thermoelectrocatalytic reduction of H_2O_2 is thermodynamically favorable.^{10, 25} The further reduction of H_2O_2 in the reaction mixture has been also reported in other *rAae*UPO-driven catalytic systems.²⁶⁻²⁷

[Manuscript, page 17, lines 1-11]

Ultraviolet photoelectron spectroscopy (UPS). UPS spectra were measured using electron spectroscopy for chemical analysis (ESCA) apparatus (Axis Supra, Kratos, UK) under an ultrahigh vacuum (base pressure $<5 \times 10^{-10}$ Torr). He I radiation source with a photon energy of 21.2 eV was used. The apparatus was calibrated using a gold standard sample stored in a load lock chamber. The sample surface was etched using Ar cluster sputter gun for 60 s before acquisition to exclude the effect of surface oxidation because the typical information depth of UPS is approximately 2–3 nm. The Fermi level (E_F) in the unit of eV with respect to the vacuum level (E_{vac}) was determined using following equation:

$$E_{\text{vac}} - E_F = -h\nu + E_{\text{cutoff}}$$

, where $h\nu$ is the energy of incident light (21.2 eV) and E_{cutoff} is the secondary electron cut-off energy.

Supplementary Figure 8. Ultraviolet photoelectron spectroscopy (UPS) spectra of as-synthesized n-type Bi_2Te_3 for (A) the valence band and (B) the secondary electron cut-off region. He I radiation source having a photon energy of 21.2 eV was used for the measurement, yielding a Fermi level energy (E_F) of 4.3 eV with respect to the vacuum level.

Supplementary Figure 9. Schematic energy diagram of as-synthesized n-type Bi₂Te₃ particles for thermoelectrocatalytic O₂ reduction to H₂O₂. The conduction band minimum potential and the electron chemical potential (i.e., the Fermi level, E_F) is more negative than the redox potential of O₂/O₂^{•-}; thus, electrons can be transferred from the surface of Bi₂Te₃ to O₂ in the electrolyte, resulting the formation of H₂O₂. The energy diagram is determined from the DRIFTS and UPS measurements.

2. Mechanism

We could reveal that the electrons in the conduction band react with O₂ during our thermoelectrocatalytic ORR, which explains the main reaction for H₂O₂ production; however, the other half-reaction to sustain the whole reaction would need more clarification. From the energy band diagram, the VBM of the Bi₂Te₃ (0.04 V vs NHE) is not sufficiently positive to oxidize water (H₂O/O₂, 1.23 V vs. NHE; *Nat. Catal.* **2020**, *3*, 96-97) or hydroxide (OH⁻/OH[•], 1.99 V vs. NHE; *Adv. Energy. Mater.* **2021**, *11*, 2003500). Therefore, it is energetically unfavorable for holes to govern the other half-reaction. The use of typical electron donors (e.g., methanol and formaldehyde) did not boost the H₂O₂ productivity of Bi₂Te₃ as well (**Supplementary Fig. 18**).

Alternatively, we speculate that the consumed electrons in Bi₂Te₃ can be replenished by accommodating oxygen into Bi₂Te₃ lattice, thus acting as the other half-reaction for ORR. According to the literature, the incorporated oxygen in Bi₂Te₃ acts as an electron donor by a complex interaction between vacancy, oxygen, and antisite defects to maintain charge neutrality of the matrix [*J. Appl. Phys.* **1962**, *33*, 2443-2450; *ACS Appl. Mater. Interfaces.* **2021**, *13*, 60216-60226]. Similar cases have been widely reported in photocatalytic reactions, where the catalyst itself participates in the half reaction [*Appl. Catal. B* **2017**, *217*, 485-493; *Angew. Chem. Int. Ed.* **2018**, *57*, 13613-13617; *Nanoscale* **2020**, *12*, 1213-1223]. We have included the result and discussion in the revised manuscript.

[Manuscript, page 11, lines 17-24]

We could reveal that the electrons in the conduction band react with O₂ during our thermoelectrocatalytic ORR, which explains the main reaction for H₂O₂ production; however, the other half-reaction to sustain the whole reaction would need more clarification. From the energy band diagram, the VBM of the Bi₂Te₃ (0.04 V vs NHE) is not sufficiently positive to oxidize water (H₂O/O₂, 1.23 V vs. NHE³⁶) or hydroxide (OH⁻/OH[•], 1.99 V vs. NHE³⁷). Therefore, it is energetically unfavorable for holes to govern the other half-reaction. The use of typical electron donors (e.g., methanol and formaldehyde) didn't boost the H₂O₂ productivity of Bi₂Te₃ as well (**Supplementary Fig. 18**).

[Manuscript, page 12, lines 1-6]

Alternatively, we speculate that the consumed electrons in Bi₂Te₃ can be replenished by accommodating oxygen into Bi₂Te₃ lattice, thus acting as another half-reaction for ORR. It is known that the incorporated oxygen in Bi₂Te₃ acts as an electron donor by a complex interaction between vacancy, oxygen, and antisite defects to maintain charge neutrality of the matrix.³⁸⁻³⁹ Similar cases have been widely reported in photocatalytic reactions, where the catalyst itself participates in the half reaction.⁴⁰⁻⁴²

Supplementary Figure 18. Effect of typical electron donors on Bi₂Te₃-driven thermoelectrocatalytic generation of H₂O₂. Reaction conditions: 5 mg mL⁻¹ Bi₂Te₃ dispersed in an O₂-purged potassium phosphate-buffered solution (KPB, 100 mM, pH 7.0) with applied temperature difference ($\Delta T = 45$ K) for 1 h. All reported values represent the mean \pm standard deviation ($n = 3$).

3. Because this research is in its infancy stage, I suggest making predictions about the best achievable performance, for example, the energy levels could be tuned in Bi₂Te₃.

Answer: We acknowledge the importance of making predictions about the best achievable performance when utilizing TEC or TEBC. As you suggested, the energy levels of VBM and CBM of Bi₂Te₃ can be tuned via Se alloying to Te sites [Research 2020, 2020, 4361703], and it would significantly change the overall energetics for the reaction, and would increase the reaction rate. However, a detailed relationship between the energy of VBM, CBM, and the reaction rate is unclear, and remains as an intriguing future work.

Nevertheless, we could determine the maximum H₂O₂ generation rate when using Bi₂Te₃ as a catalyst based on the TEPOC model. For example, the maximum S value of Bi₂Te₃ is known to be $-250 \mu\text{V K}^{-1}$ [Adv. Electron. Mater. 2019, 5, 1800904]. We expect that the generation rate might be 10.9 times faster ($\sim 4.03 \mu\text{M min}^{-1}$, when r_0 is $0.37 \mu\text{M min}^{-1}$) than that without the TEPOC effect if we utilize Bi₂Te₃ particles with a S value of $-250 \mu\text{V K}^{-1}$. We have included above discussion in the revised manuscript accordingly.

[Manuscript, page 12, line 22 – page 13, line 6].

For example, it is known that the maximum S value of Bi₂Te₃ can be as high as $-250 \mu\text{V K}^{-1}$ via careful tuning of the Fermi level.⁹ Therefore, based on the TEPOC model, we expect that the best achievable H₂O₂ generation rate of Bi₂Te₃ might be 10.9 times faster ($\sim 4.03 \mu\text{M min}^{-1}$, when r_0 is $0.37 \mu\text{M min}^{-1}$) than that without the TEPOC effect if we utilize Bi₂Te₃ particles with a S value of $-250 \mu\text{V K}^{-1}$. Moreover, we expect band engineering would be an ideal strategy to synergistically improve the performance of the reaction. It is known that the band edge positions of both valence band and conduction band can be tuned by alloying Se to Te sites.⁴⁵ Together with tuning carrier concentration, one might achieve an optimal alignment of the Fermi level and reaction potential.

4. It would be great to compare this TEPOC with other related technologies, even if current productivity is still low.

Answer: In response to the Referee's comment, we have compared total turnover number of *rAaeUPO* ($TTN_{rAaeUPO}$) of the $\text{Bi}_2\text{Te}_3/rAaeUPO$ couple with those of other state-of-the-art *rAaeUPO*-driven biocatalytic systems, and plotted as **Supplementary Figure 19** in the revised manuscript.

[Manuscript, page 12, lines 9-11]

The highest $TTN_{rAaeUPO}$ of the TEBC measured in this study is comparable to, or even exceeds that of other state-of-the-art *rAaeUPO*-driven biocatalytic systems (**Supplementary Fig. 19**).

Supplementary Figure 19. Comparison of total turnover number of *rAaeUPO* ($TTN_{rAaeUPO}$) values achieved for UPO-catalyzed selective oxyfunctionalization reactions in this study and up-to-date *rAaeUPO*-driven biocatalytic systems. The catalysts include gold-loaded TiO_2 (Au-TiO_2)^{1,2}, Mo-doped BiVO_4 (Mo:BiVO_4)³, BiOCl ⁴, graphitic carbon nitride ($\text{g-C}_3\text{N}_4$)⁵, TiO_2 ⁶, AuPd decorated TiO_2 (AuPd/TiO_2)⁷, phenosafranine⁸, flavin mononucleotide (FMN)^{8,9}.

[Supporting information, page 19, references]

References

- (1) Zhang, W.; Burek, B. O.; Fernandez-Fueyo, E.; Alcalde, M.; Bloh, J. Z.; Hollmann, F. Selective Activation of C-H Bonds in a Cascade Process Combining Photochemistry and Biocatalysis. *Angew. Chem. Int. Ed.* **2017**, *56*, 15451–15455.
- (2) Zhang, W.; Fernández-Fueyo, E.; Ni, Y.; van Schie, M. M. C. H.; Gacs, J.; Renirie, R.; Wever, R.; Mutti, F. G.; Rother, D.; Alcalde, M.; Hollmann, F. Selective aerobic oxidation reactions using a combination of photocatalytic water oxidation and enzymatic oxyfunctionalizations. *Nat. Catal.* **2018**, *10*, 55-62.
- (3) Choi, D. S.; Kim, J.; Hollmann, F.; Park, C. B. Solar-Assisted eBiorefinery: Photoelectrochemical Pairing of Oxyfunctionalization and Hydrogenation Reactions. *Angew. Chem. Int. Ed.* **2020**, *59*, 15886-15890.
- (4) Yoon, J.; Kim, J.; Tieves, F.; Zhang, W.; Alcalde, M.; Hollmann, F.; Park, C. B. Piezobiocatalysis: Ultrasound-Driven Enzymatic Oxyfunctionalization of C-H Bonds. *ACS Catal.* **2020**, *10*, 5236-5242.
- (5) van Schie, M. M. C. H.; Zhang, W.; Tieves, F.; Choi, D. S.; Park, C. B.; Burek, B. O.; Bloh, J. Z.; Arends, I. W. C. E.; Paul, C. E.; Alcalde, M.; Hollmann, F. Cascading $\text{g-C}_3\text{N}_4$ and Peroxygenases for Selective Oxyfunctionalization Reactions. *ACS Catal.* **2019**, *9*, 7409–7417.
- (6) Burek, B. O.; de Boer, S. R.; Tieves, F.; Zhang, W.; van Schie, M. M. C. H.; Bormann, S.; Alcalde, M.; Holtmann, D.; Hollmann, F.; Bahnemann, D. W.; Bloh, J. Z. Photoenzymatic Hydroxylation of Ethylbenzene

Catalyzed by Unspecific Peroxygenase: Origin of Enzyme Inactivation and the Impact of Light Intensity and Temperature. *ChemCatChem* **2019**, *11*, 3093-3100.

(7) Freakley, S. J.; Kochius, S.; van Marwijk, J.; Fenner, C.; Lewis, R. J.; Baldenius, K.; Marais, S. S.; Opperman, D. J.; Harrison, S. T. L.; Alcalde, M.; Smit, M. S.; Hutchings, G. J. A chemo-enzymatic oxidation cascade to activate C–H bonds with in situ generated H₂O₂. *Nat. Commun.* **2019**, *10*:4178.

(8) Willot, S. J. P.; Fernández-Fueyo, E.; Tieves, F.; Pesic, M.; Alcalde, M.; Arends, I. W. C. E.; Park, C. B.; Hollmann, F. Expanding the Spectrum of Light-Driven Peroxygenase Reactions. *ACS Catal.* **2019**, *9*, 890-894.

(9) Al-Shameri, A.; Willot, S. J. P.; Paul, C. E.; Hollmann, F.; Lauterbach, L. H₂ as a fuel for flavin- and H₂O₂-dependent biocatalytic reactions. *ChemComm.* **2020**, *56*, 9667-9670.

REVIEWERS' COMMENTS

Reviewer #1 (Remarks to the Author):

This is a much improved version, all of issues have been addressed well. Agree for acceptance for publication.

Reviewer #2 (Remarks to the Author):

The author has addressed most of the comments, including energy diagram, mechanism, and comparison with other works. This work can be published in its current form.